# Structure of the *Pseudomonas aeruginosa* PAO1 Type IV pilus

**Hannah Ochner, Jan Böhning, Zhexin Wang, Abul K. Tarafder, Ido Caspy
Tanmay A. M. Bharat** [ID]*

Structural Studies Division, MRC Laboratory of Molecular Biology, Francis Crick Avenue, Cambridge, United Kingdom

* tbharat@mrc-lmb.cam.ac.uk

**Data Availability Statement:** The EM density maps, atomic models and validation reports for the P. aeruginosa Type IV pilus structure presented here are available via PDB (9EWX) and EMDB

## Abstract

Type IV pili (T4Ps) are abundant in many bacterial and archaeal species, where they play important roles in both surface sensing and twitching motility, with implications for adhesion, biofilm formation and pathogenicity. While Type IV pilus (T4P) structures from other organisms have been previously solved, a high-resolution structure of the native, fully assembled T4P of *Pseudomonas aeruginosa*, a major human pathogen, would be valuable in a drug discovery context. Here, we report a 3.2 Å-resolution structure of the *P. aeruginosa* PAO1 T4P determined by electron cryomicroscopy (cryo-EM). PilA subunits constituting the T4P exhibit a classical pilin fold featuring an extended N-terminal α-helix linked to a C-terminal globular β-sheet-containing domain, which are packed tightly along the pilus, in line with models derived from previous cryo-EM data of the *P. aeruginosa* PAK strain. The N-terminal helices constitute the pilus core where they stabilise the tubular assembly via hydrophobic interactions. The α-helical core of the pilus is surrounded by the C-terminal globular domain of PilA that coats the outer surface of the pilus, mediating interactions with the surrounding environment. Comparison of the *P. aeruginosa* PAO1 T4P with T4P structures from other organisms, both at the level of the pilin subunits and the fully assembled pili, confirms previously described common architectural principles whilst highlighting key differences between members of this abundant class of prokaryotic filaments. This study provides a structural framework for understanding the molecular and cell biology of these important cellular appendages mediating interaction of prokaryotes to surfaces.

## Author summary

*Pseudomonas aeruginosa* is an important human pathogen that poses a critical problem in hospital settings, causing widespread antibiotic-resistant infections. *P. aeruginosa* evades antibiotic treatments by forming biofilms, i.e. surface-attached, multicellular communities of bacterial cells. To facilitate the transition from a unicellular planktonic to a multicellular biofilm state, *P. aeruginosa* expresses a variety of filamentous adhesins and pili on its outer surface which mediate surface sensing, surface attachment, as well as bacterial mobility. Here, we investigate Type IV pili (T4P), which play a key role in both surface

(EMD-50025). https://www.ebi.ac.uk/emdb/EMD-50025 https://www.rcsb.org/structure/9EWX.

**Funding:** This work was supported by the Medical Research Council, as part of United Kingdom Research and Innovation (also known as UK Research and Innovation) [Programme MC_UP_1201/31 to T.A.M.B.]. T.A.M.B. would like to thank EPSRC (Grant EP/V026623/1), the European Molecular Biology Organization, the Wellcome Trust (Grant 225317/Z/22/Z), the Leverhulme Trust, and the Lister Institute for Preventative Medicine for support. H.O. and I.C. were supported by EMBO Postdoctoral Fellowships (ALTF 1076-2023 to H.O. and ALTF 92-2022 to I.C.). T.A.M.B., J.B. and A.K.T. were supported through MC_UP_1201/31 and Z.W. was supported by EP/V026623/1. The funders had no role in study design, data collection and analysis, decision to publish, or preparation of the manuscript.

**Competing interests:** The authors have declared that no competing interests exist.

sensing and twitching motility in many bacteria. Using electron cryomicroscopy (cryo-EM), we determined a high-resolution structure of the Type IV pilus of the *P. aeruginosa* PAO1 strain, which reveals detailed interactions between subunits of the pilus and allows comparison with T4P structures of other organisms. This provides a structural framework for understanding these important cellular appendages, which is highly relevant for the development of strategies to combat antibiotic-resistant infections.

## Introduction

Surface sensing and adhesion are crucial for bacteria and archaea to colonise new environments [1,2], because these processes help microbes invade hosts, establish biofilms and form connections in microbiomes [3–6]. These sensing and adhesion processes are often mediated by proteinaceous, filamentous surface structures known as pili or fimbriae [7–10]. Bacterial surface pili are assembled by repeated interactions of monomeric protein subunits called pilins, leading to the formation of a filamentous structure that protrudes from the cell and binds to surfaces [11,12]. Although there is a bewildering variety of pili present on bacteria, the Type IV pili (T4Ps) of Gram-negative bacteria are profoundly involved in surface sensing and initial attachment of bacteria to surfaces [13]. Initial attachment to surfaces is not only the first step of biofilm formation [14], but is also important in pathogenicity and the establishment of infections [15]. T4P biogenesis employs a multi-component membrane complex to transport pilin subunits into the growing pilus, extruding the pilus across the periplasm and through an outer membrane secretin channel for surface display [16,17]. T4Ps are dynamic filaments that can extend and retract [18,19], allowing cells to rapidly respond to a changing environment, particularly when encountering a favourable surface suitable for colonisation.

*Pseudomonas aeruginosa* is an important Gram-negative human pathogen that poses a critical problem in hospital settings [20], causing widespread antibiotic-resistant infections [21]. *P. aeruginosa* evades antibiotic treatments by forming biofilms [22,23], which makes clearing bacterial infections extremely challenging. Biofilm formation in *P. aeruginosa* is facilitated by the expression of a variety of filamentous adhesins [6,24] and pili [13,14] on its outer surface. As *P. aeruginosa* T4Ps mediate surface sensing [14], the first step of biofilm formation, they play a key role in allowing *P. aeruginosa* to transition from a unicellular planktonic to a multicellular biofilm state. Understanding the structure of the *P. aeruginosa* Type IV pilus is thus highly relevant for the development of strategies to combat antibiotic-resistant infections.

On the structural level, several studies have reported the structures of the filamentous T4Ps from a variety of prokaryotes, including *Neisseria gonorrhoeae*, *Escherichia coli*, *Myxococcus xanthus*, *Pyrobaculum arsenaticum* and *Thermus thermophilus* [25–31] using electron cryomicroscopy (cryo-EM) single-particle analysis. Other studies have elucidated the organisation of the secretion machinery of T4Ps in different bacterial species using electron cryotomography (cryo-ET), which features ring-like densities corresponding to the secretin channel, assembly and retraction complexes [32–34]. Cryo-EM analysis of the *P. aeruginosa* secretin, a homomultimer of PilQ, revealed a channel with a gate at the periplasmic face, resembling the shape of secretins involved in type II and type III secretion [35].

Previous studies have reported an X-ray crystal structure of the pilin monomer from *P. aeruginosa* strain K (PAK) [36], which was later docked into an 8 Å resolution cryo-EM map of the *P. aeruginosa* PAK pilus [26] to propose a pseudo-atomic model of the PAK pilus. Despite this, a high-resolution structure of the filamentous T4P from *P. aeruginosa* has remained

elusive, and there is little direct structural information available for the T4P of the highly studied model strain PAO1.

In this study, we describe a 3.2 Å-resolution cryo-EM structure of the fully assembled *P. aeruginosa* T4P from the model PAO1 strain, the highest-resolution structure of a *P. aeruginosa* T4P using cryo-EM to date. Our structure allows us to derive an atomic model for the pilus, which provides detailed molecular insights into the assembly and three-dimensional arrangement of this important appendage. Furthermore, our structural data allows a detailed comparison of this pilus with other solved T4P structures from multiple prokaryotic species, which is relevant for placing past and future structural and cell biology studies of this abundant pilus into context.

## Results

### Purification of natively assembled T4Ps

To study the structure of the *P. aeruginosa* T4P, we used a PAO1 strain with a deletion of the retraction ATPase *pilT* [13,19,37,38] to maximise pilus decoration on the cell surface. Cells from this strain were deposited on electron microscopy grids and visualised with cryo-ET (Fig 1a). Multiple T4Ps were observed protruding from the outer membrane of these PAO1 Δ*pilT* cells, with the same overall morphological appearance as T4Ps described in other bacteria and PAO1 wild-type strains [34].

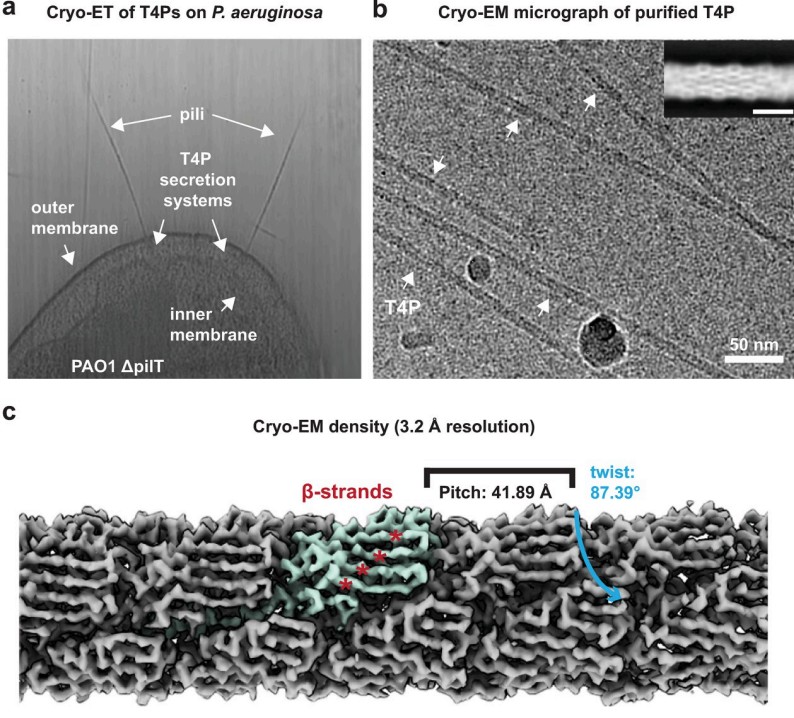

**Fig 1. Cellular arrangement and structure of the *P. aeruginosa* PAO1 T4P. a** Slice through a tomogram of a *P. aeruginosa* PAO1 Δ*pilT* cell with T4Ps protruding from the cell. Cell membrane, T4Ps, as well as T4P secretion machinery, which assembles the T4Ps, are marked by arrows. **b** Cryo-EM micrograph of T4Ps purified from *P. aeruginosa* PAO1 Δ*pilT* cells, with T4Ps labelled by white arrows. The inset shows a representative 2D class average (scale bar 50 Å). **c** Cryo-EM density map of the assembled T4P reconstructed in RELION 4.0 with one of the PilA subunits highlighted in turquoise. The map resolution of 3.2 Å shows a clear separation of the β-strands in each of the PilA subunits (marked by red asterisks). TheT4P has a helical rise of 10.17 Å, a helical twist of 87.39˚ and thus a helical pitch of 41.89 Å.

The increased occurrence of T4Ps on the cell surface of the PAO1 Δ*pilT* strain facilitated pilus isolation for structural analysis. We adapted a previously described methodology for purifying T4Ps [39] to isolate *P. aeruginosa* T4Ps. In our preparations, along with the T4Ps sheared from cells, bacterial flagella were present as contaminants (S1 Fig). For cryo-EM analysis, the contaminant flagella did not pose an insurmountable problem because of the large difference in diameter of the two filaments (flagella: ~170 Å [40], T4P: 51 Å), which allowed us to clearly distinguish them in micrographs. Peptide fingerprinting mass spectrometry confirmed the presence of *P. aeruginosa* PilA, the Type IVa pilin [41,42] protein constituting the T4P subunits, in our preparation with 73% coverage of the mature PilA sequence (S1 Fig).

## Structure of the *P. aeruginosa* PAO1 T4P using cryo-EM helical reconstruction

Once the presence of natively assembled T4Ps was confirmed in our sample, we proceeded to acquire cryo-EM data on this specimen (Fig 1b), which allowed us to produce two-dimensional (2D) class averages of the *P. aeruginosa* PAO1 T4P (Fig 1b, inset). The 2D class averages showed a tubular assembly with a diameter of approximately 51 Å.

Next, starting from previously reported helical symmetry parameters of the *P. aeruginosa* PAK strain T4P [26], whose PilA subunit has 67% sequence identity with the PAO1 PilA, we performed repeated three-dimensional classification and refinement (S2 Fig). Particles from classes not showing densities that resembled protein subunits were discarded. The remaining classes, which exhibited significantly improved image processing statistics and cryo-EM densities, were retained for further processing. The helical symmetry of these classes was applied to the remaining dataset in subsequent refinements, which allowed us to perform helical reconstruction and obtain a 3.2 Å-resolution cryo-EM map with clearly resolved side chain densities (Figs 1c and S3–S5), from which an atomic model of the T4P could be built (S1 Table and S1 Movie and Figs 2 and S3–S4).

The atomic model confirms a tubular arrangement of the *P. aeruginosa* PAO1 T4P (Fig 2a), with a helical rise of 10.17 Å and a right-handed twist of 87.4° per subunit, resulting in an overall helical pitch of 41.9 Å (Figs 1c and 2a). In agreement with T4Ps of other species as well as the structure of the *P. aeruginosa* PAK strain T4P [26], the N-terminal α-helix of each PilA subunit is stacked in the centre of the tubular T4P to form the filament (Fig 2a and 2b). This N-terminal helix is connected through a short linker to the helical part of the C-terminal globular pilin domain. The globular C-terminal pilin domain contains an α-helix and a β-sheet consisting of four β-strands that face the exterior of the T4P and are thus exposed to the solvent (Fig 2c). These β-strands are stabilised by the globular domain α-helix via hydrophobic interactions along the T4P tube (Figs 2c and S6). Two additional structural motifs common to pilins can be identified within the globular domain, namely the αβ-loop linking the α-helix and the β-sheet and the D-region at the C-terminus [43,44] (marked in Fig 2c).

The innermost part of the T4P is stabilised by hydrophobic interactions of the N-terminal α-helices of PilA, which contain a row of hydrophobic residues (amino acid residues 1–16, where residue 1 is the N-terminus of the mature pilin), forming the core of the T4P in a staggered arrangement (Fig 2d). These N-terminal helices in turn interact with the helices of the globular domain through further hydrophobic interactions (Fig 2b and 2d). Moreover, ionic interactions and hydrogen bonds, mainly involving the loops in the pilin globular domains, stabilise the interfaces among multiple pilin subunits along the assembled pilus (Fig 3). For example, the Q23 and Q25 residues of every pilin subunit N interact with the β3-β4-loop of subunit N-4 and the αβ-loop of subunit N-3 (Fig 3c). Similarly, residue K139 of every subunit N interacts with the β2-β3-loop of subunit N+1 and the αβ-loop of subunit N-3 (Fig 3d).

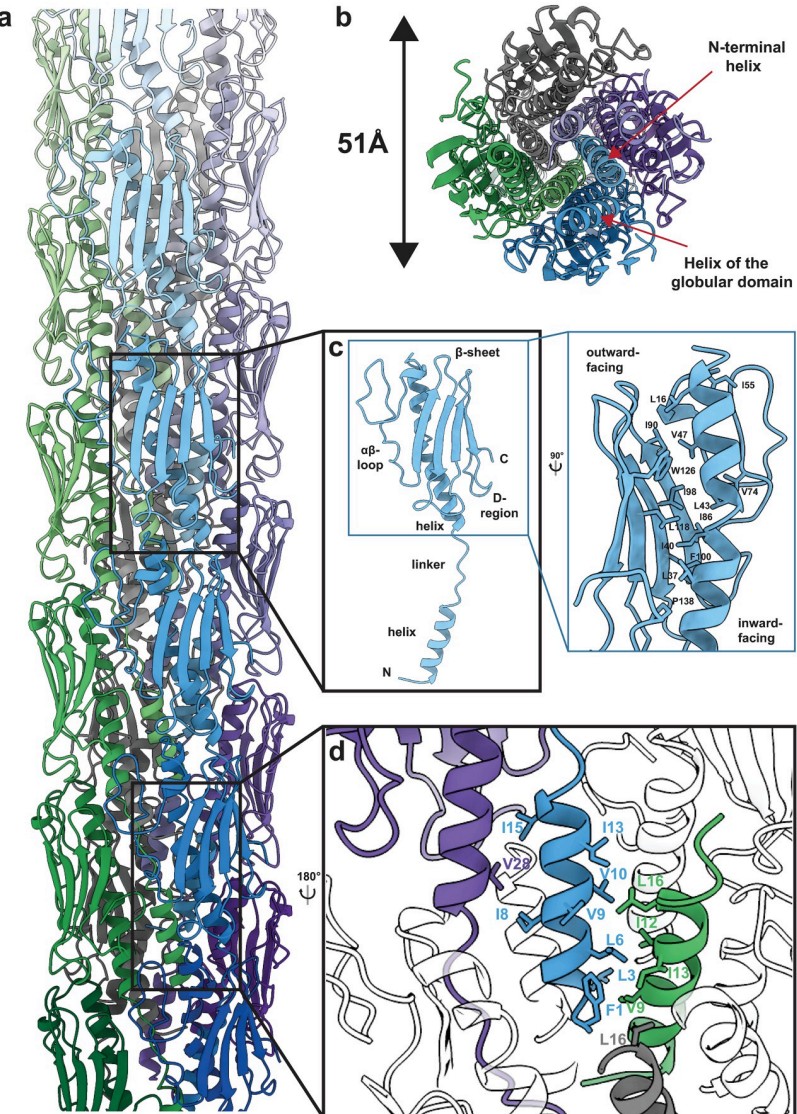

**Fig 2. Structural features of the *P. aeruginosa* PAO1 T4P. a** Model of the assembled *P. aeruginosa* PAO1 T4P. **b** Cross section through the model showing the tightly packed N-terminal α-helices in the centre of the pilus, which in turn interact with helices in the globular domain of other pilin subunits (marked by red arrows). **c** Structure of an individual PilA subunit composing the pilus. The PilA subunit features an N-terminal α-helix and a globular domain consisting of an α-helical segment connected to a β-sheet by a linker called αβ-loop. At the C-terminus, an additional loop, called D-region, can be identified. The two helices (N-terminal and globular domain α-helices) are coupled by a 'melted' linker region. N- and C-termini are marked N and C, respectively. The inset highlights the strong hydrophobic interactions between the α-helix and β-sheet in the globular domain, which stabilise the pilin structure. **d** Hydrophobic subunit-subunit interactions in the core of the T4P are mediated by the PilA α-helices.

These molecular bridges on the periphery together with the packed N-terminal helices in the centre result in an ordered, tubular pilus (Figs 2a and 3a). While amino acid side chains were not resolved in previous cryo-EM reconstructions of the *P. aeruginosa* T4P, they can be resolved unambiguously in our map.

In addition to the side chain information directly resolved in our 3.2 Å-resolution cryo-EM map, a direct comparison with the previously published model of the *P. aeruginosa* PAK strain

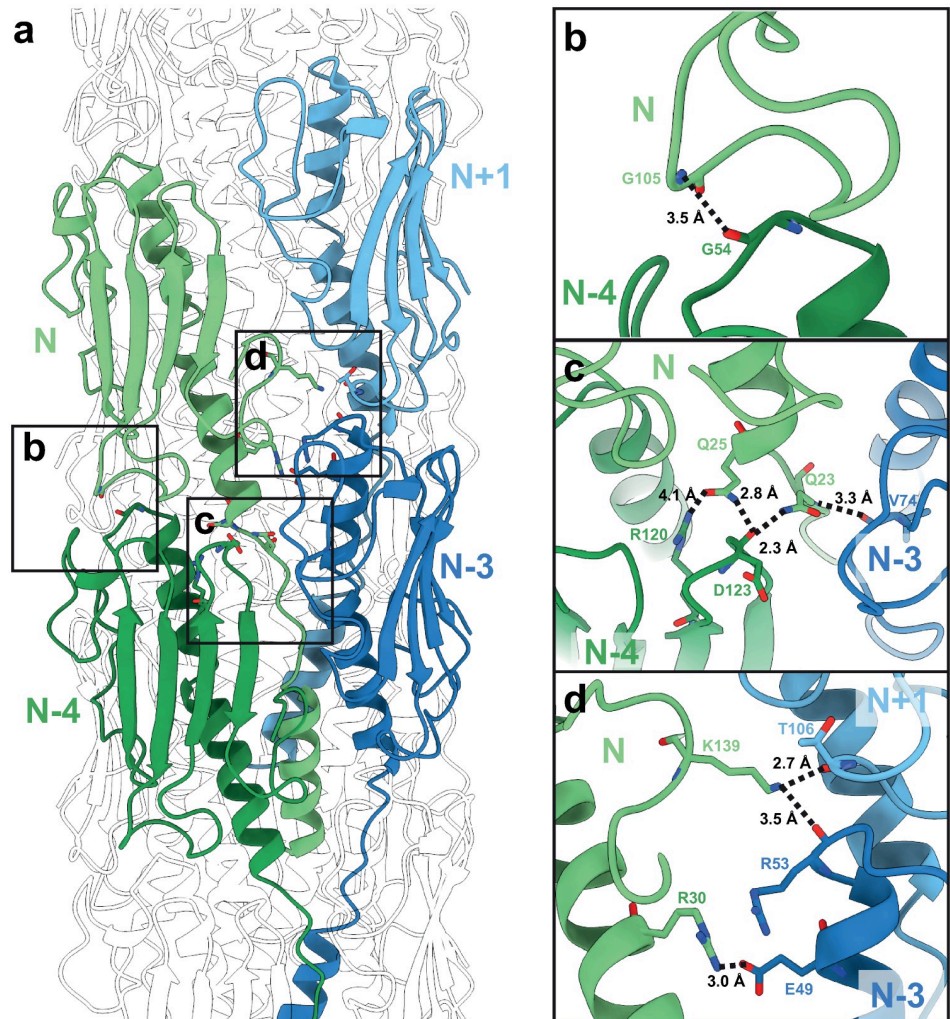

**Fig 3. Interactions between pilin subunits at the periphery of the pilus. a** Structural model of four neighbouring pilin subunits (labelled N, N+1, N-3, N-4 based on the helical arrangement). **b** Interaction between the N and the N-4 subunits mediated through loops resolved in our 3.2 Å-resolution cryo-EM map. **c** Interactions at the interface among subunits N, N-3 and N-4 via side chains resolved in our cryo-EM map. **d** Interactions at the interface between subunits N, N-3 and N+1, via side chains resolved in our cryo-EM map.

pilus, which was derived from fitting an X-ray crystallographic model into an 8 Å-resolution cryo-EM map [26], shows differences even on the overall level of secondary structural elements (S5 Fig). The comparison highlights different relative orientations between the pilin globular domain and the N-terminal helix (S5c and S5d Fig), as well as significant differences in the size of αβ-loop (S5e Fig). Furthermore, additional secondary structure formation within the loop region is observed (residues 56–58, 60–61, 72–73; S5e Fig, marked by red asterisks), as well as differences in length of the β-strands, all of which are shorter in PAK than in PAO1, with the largest difference in the β4-strand (residues 126–130 (PAO1) vs residues 130–131 (PAK), S5f Fig, marked by a red asterisk). These discrepancies between the two structural models likely originate from a combination of differences between the strains (S5a Fig), different experimental conditions between cryo-EM and crystallographic structural analysis, as well as the higher accuracy of the 3.2 Å-resolution cryo-EM map that we report in this study.

Previous work has shown that five groups of T4Ps are present in *P. aeruginosa* strains, with each strain expressing a single group. Two of these groups are glycosylated (Group 1 and Group 4), which are modifications that help bacteria escape phages that use the T4P as an entry receptor [45–47]. *P. aeruginosa* PAO1, however, is a strain that encodes a Group 2 pilin that is not glycosylated, in good agreement with our cryo-EM density which did not show any densities that are not accounted for by the polypeptide chain itself (S1 Movie). Finally, modelling the *P. aeruginosa* PAO1 T4P alongside the PilQ secretin EM density [35] (S7 Fig) shows that the T4P fits well into the PilQ lumen, which is consistent with the role of the secretin in T4P assembly.

## Comparison of T4P structures

The PilA subunit of the *P. aeruginosa* PAO1 T4P adopts a classical pilin fold, as demonstrated by comparison with pilin subunits from other organisms (Fig 4a–4g). While Type IV pilins generally consist of a globular domain and an N-terminal helix, the globular domain is considerably less conserved than the N-terminal helix (Fig 4h), which is a major mediator of subunit-subunit interactions (Fig 2d), stabilising the core of the pilus. In addition, the C-terminal region of the *P. aeruginosa* PilA contains two buried cysteine residues (C128, C141) in close proximity, which form a disulfide bond resolved in our map, constituting a structural feature present in many known T4P structures [48] (S8 Fig).

While in X-ray structures of pilins, including that of the *P. aeruginosa* PAK strain, the experimental conditions lead to a relaxation of the N-terminal part of the pilin into a continuous helix spanning the full length of the subunit [49,50] (S9 Fig), many cryo-EM structures of pilins determined within the full pilus architecture—including our 3.2 Å-resolution structure —feature a non-helical but well-resolved linker segment (Figs 2c and S4b) containing a prominent helix-breaking proline residue (P22, S4b Fig), which divides the α-helical part of PilA into an N-terminal helix (residues 1–16) and a distinct α-helix that is part of the globular domain (residues 25–52). This so-called 'melting' of the middle of the helical region is important for the assembly of the pilus [26] and is broadly conserved among bacterial pilins (Fig 4a–4f), while more divergent organisms, such as the archaeon *Pyrobaculum arsenaticum* (Fig 4g), exhibit a more extensive N-terminal helix without melted segments. Several studies have indicated that the melted segments allow the T4P to stretch considerably when subjected to force, thus suggesting a functional role of this structural feature [44,51].

When comparing the Type IV pilins from various prokaryotic organisms at the domain level, several interesting features are discerned. While the globular domains are similar at an organisational level, the presence of the 'melted' linker segment results in a different positioning of the N-terminal helix relative to the globular domain, influencing the helical symmetry parameters and thus overall pilus architecture (S2 Table). This becomes especially clear when aligning the pilin structures as shown in Fig 4a–4g, where each of the pilin models is presented in an orientation resulting from structural alignment to the *P. aeruginosa* PAO1 pilin structure (Fig 4a). To quantify these similarities and differences, we calculated RMSD values comparing the *P. aeruginosa* PAO1 model to all the other models presented here (S3 Table).

A comparison of T4P structures originating from different organisms is not only of interest on the level of the pilin structure, but also when considering the overall architecture of the fully assembled pili (Fig 5). All T4Ps present an overall straight, tubular geometry, with two major types of helical parameters observed for bacterial (see S2 Table) and archaeal pili (with exception of the *Caulobacter crescentus* Tad pili), described in detail previously [52]. Despite the relatively similar helical symmetry amongst bacterial T4Ps, the pilus structures differ significantly in diameter, ranging from 51–80 Å. The *P. aeruginosa* T4P shows the smallest

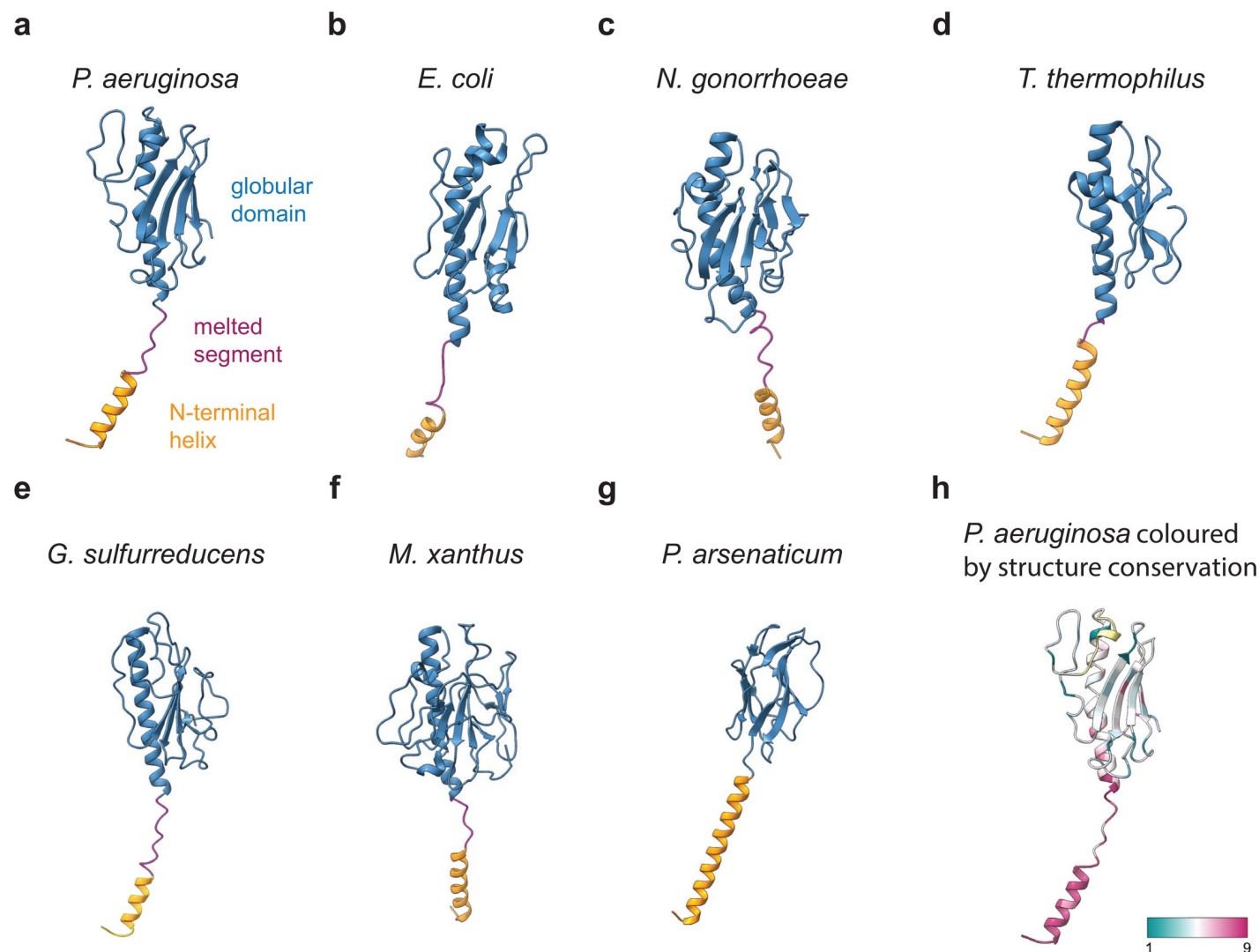

**Fig 4. Comparison of Type IV pilin structures from different organisms. a-g** Aligned structures of Type IV pilins from different organisms: *P. aeruginosa* PAO1 (this study, PDB: 9EWX) (**a**), *E. coli* (PDB: 6GV9) (**b**), *N. gonorrhoeae* (PDB: 5VXX) (**c**), *T. thermophilus* (PDB: 6XXD) (**d**), *G. sulfurreducens* (PDB: 6VK9) (**e**), *M. xanthus* (PDB: 8TJ2) (**f**), *P. arsenaticum* (PDB: 6W8U) (**g**). The pilins are coloured by domain: N-terminal helix (orange), melted segment (magenta), globular domain (blue). The pilin structures have been aligned to the *P. aeruginosa* pilin using the ChimeraX *matchmaker* function. The resulting orientations show the structural diversity between the pilins introduced by the presence of the melted linker region featured by most pilin structures. **h** *P. aeruginosa* PilA pilin ribbon diagram coloured based on structure conservation score from highly conserved (9) to low conservation (1) calculated by comparing the pilin structures presented in **a-g** by sequence alignment using ConSurf [87–89]. While the N-terminal α-helix is highly conserved, the globular domain of the pilin exhibits low conservation.

diameter (51Å) by a large margin (S2 Table). This comparatively small diameter is remarkable when combined with the observation that the *P. aeruginosa* T4P does not present flexible loops on the outer surface of the pilus, which are observed prominently in the structures of other pili (Fig 5, especially 5c-e, g, marked by red arrows). As flexible residues could potentially be targets for extracellular proteases, a lack of such residues might make the *P. aeruginosa* T4P more resistant to proteolysis, suggesting an interesting avenue of research to explore.

A comparison of surface electrostatics and hydrophobicity properties across the T4P species (S10 and S11 Figs) shows that all T4Ps feature a hydrophilic outer surface and a hydrophobic core. While the surface charge is predominantly negative or neutral, all T4Ps feature regions of

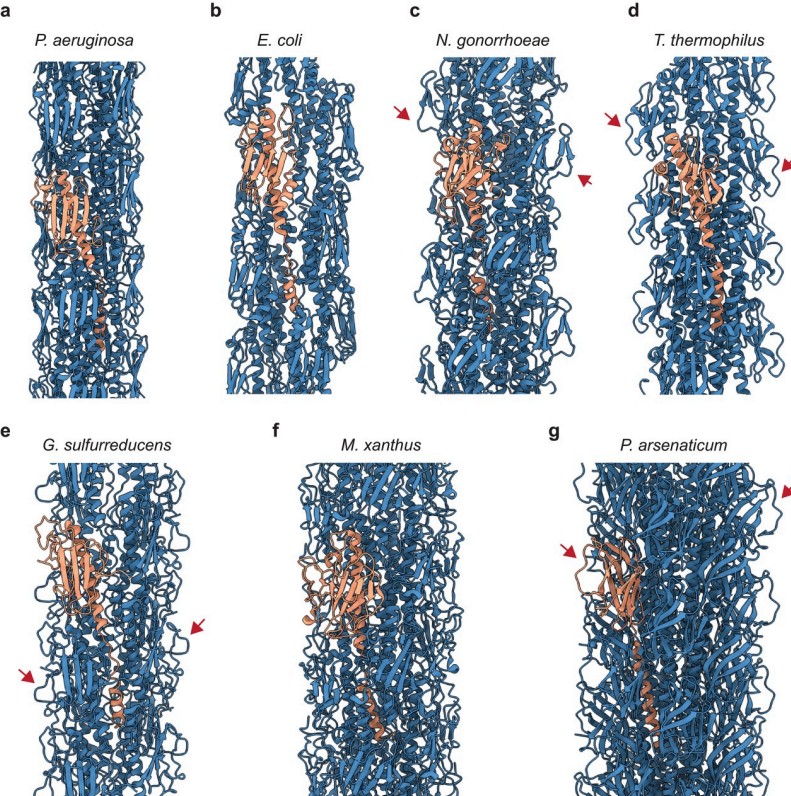

**Fig 5. Comparison of T4P architectures in different organisms. a-g** Models of T4P structures from different organisms with one pilin subunit highlighted in orange for each structure: *P. aeruginosa* PAO1 (this study, PDB: 9EWX) (**a**), *E. coli* (PDB: 6GV9) (**b**), *N. gonorrhoeae* (PDB: 5VXX) (**c**), *T. thermophilus* (PDB: 6XXD) (**d**), *G. sulfurreducens* (PDB: 6VK9) (**e**), *M. xanthus* (PDB: 8TJ2) (**f**), *P. arsenaticum* (PDB: 6W8U) (**g**). Flexible loops on the pilus surface are marked by red arrows.

positive charge of varying size (S10 Fig). As patches of positive charge may be related to the DNA binding capability of T4Ps reported in previous studies [53,54], the differences across species might indicate some variation in DNA binding capacity in the different species; however, DNA binding mechanisms of T4Ps are not restricted to the major pilins, but can be mediated by other T4P components [55].

## Discussion

In this study, we present the structure of the T4P from the important human pathogen *P. aeruginosa*. Past studies on the *P. aeruginosa* T4P from a different strain (PAK strain), had reported an 8 Å-resolution cryo-EM map, from which an atomic model could not be directly derived, but indirectly inferred by fitting the X-ray crystallographic structure of the PAK pilin into the map [26]. With improved methods for cryo-EM [56], image analysis [56–62] and helical three-dimensional classification [63–67], we were able to directly deduce the helical symmetry of the *P. aeruginosa* T4P PAO1 strain and resolve its structure to 3.2 Å-resolution, where the polypeptide backbone and side chains could be unambiguously fitted (Figs 1, 2, and S4). We note that concurrent with the submission of this manuscript, a 3.6 Å-resolution cryo-EM structure of *P. aeruginosa* T4P, which largely agrees with and confirms our structural data, was reported

in a study where the interaction of the T4P with single-stranded RNA phage PP7 [68] was probed.

Our structure reveals that the *P. aeruginosa* PilA protein adopts a classical pilin fold, sharing similarities with other pilins across the prokaryotic tree of life (Fig 4). The N-terminal α-helix of PilA extends away from the body of the pilin globular domain to stack with other pilin sub-units, mediating major subunit-subunit interactions within the pilus. Such a 'hydrophobic arm'-like element is also commonly found in other pili where it similarly provides extensive contact with hydrophobic regions of other subunits, for example in filaments employing donor-strand exchange [63,69,70]. A unique feature of T4Ps is the simultaneous interaction of one pilin subunit region with various regions of several other subunits within the pilus (Figs 2 and 3). As a result of the interactions discussed above, PilA monomers of *P. aeruginosa* PAO1 stack in three-dimensional space to form a tubular T4P, resembling pili from other species (Fig 5). While the overall architecture of the T4Ps among different organisms is similar, they differ significantly in diameter and in the details of their outer surface structure.

The base of the T4P filament solved in this study interacts with T4P secretion machinery in the periplasmic space of *P. aeruginosa* cells. Although the secretion machineries of other bacteria have been studied structurally [32–34], characterisation of the T4P secretion machinery of *P. aeruginosa* at the atomic level remains an open avenue of research. The other end of the pilus contains an important *P. aeruginosa* adhesin called PilY1 [71]. While PilY1 should be retained during purification, our cryo-EM data of pili tips exhibited a strong structural heterogeneity, which hindered convergence to a convincing density for the PilY1-bound tip. Therefore, the question of how PilY1 interacts with and decorates the pilus tip remains unresolved. Further studies on native cellular systems, i.e. cells and biofilms, will be required to determine the exact mode of interaction between different T4P components [72].

Our structure will serve as a reference for future molecular and cellular studies on this T4P from a major human pathogen. Future functional experimental research needs to be performed on the *P. aeruginosa* T4P to understand its precise mechanistic role in mediating surface adhesion, motility and human infection.

## Materials and methods

### Isolation of T4Ps from *P. aeruginosa*

T4Ps were isolated and purified following a protocol adapted from Rivera *et al* [39]. Briefly, *P. aeruginosa* PAO1 Δ*pilT* (kind gift from Urs Jenal, University of Basel) was streaked from a glycerol stock on Lysogeny broth (LB)-agar plates and incubated overnight at 37˚C. Four single colonies from these plates were streaked onto new LB-agar plates and grown overnight at 37˚C. Cells were scraped from these plates and resuspended in LB media to obtain a suspension with an optical density ($OD_{600}$) of 8.0. 50 μl of this suspension was spread onto 30 LB-agar plates, which were incubated overnight at 37˚C until bacterial lawns were formed. Lawns were harvested in 5 ml ice-cold buffer A per plate (150 mM ethanolamine pH 10.5, 1 mM dithiothreitol) and incubated for 1 hour at 4˚C with stirring. To shear T4Ps from cells, samples were transferred to 50 ml Oakridge centrifuge tubes (Beckman Coulter) and vortexed three times at maximum strength for a duration of 1 minute at a time with 2-minute intervals on ice in between. To remove cells, samples were centrifuged at 15,000 relative centrifugal force (rcf) for 30 minutes at 4˚C, supernatants collected and centrifuged again at 15,000 rcf for 10 minutes at 4˚C. The supernatants from the second centrifugation step were then dialysed overnight at 4˚C against ice-cold buffer B (50 mM Tris pH 7.5, 150 mM NaCl) in SnakeSkin dialysis tubing (Molecular weight cutoff 3 kDa, Thermo Fisher Scientific) until the sample reached pH 7.5, which was confirmed using pH strips. To harvest pili, the sample was

centrifuged at 20,000 rcf for 40 minutes at 4˚C. The resulting pellet containing T4Ps was resuspended in buffer B. T4Ps were further purified by adjusting the solution to 0.5 M NaCl, followed by T4P precipitation by addition of 10% (w/v) polyethylene glycol 6000 (PEG 6000, Merck) and incubation overnight at 4˚C. Precipitated T4Ps were pelleted by centrifugation at 12,000 rcf for 30 minutes at 4˚C. The resulting pellet was resuspended in buffer B and dialysed overnight against buffer B at 4˚C to remove PEG 6000. The presence of T4Ps was verified by SDS-PAGE followed by Coomassie visualisation of proteins and mass spectrometry (MRC-LMB Mass Spectrometry Facility, see S1 Fig).

## Cryo-EM and cryo-ET sample preparation

For cryo-EM grid preparation of purified T4Ps, 2.5 µl of the sample was applied to a freshly glow-discharged Quantifoil R 3.5/1 Cu/Rh 200 mesh grid and plunge-frozen into liquid ethane using a Vitrobot Mark IV (ThermoFisher) at 100% humidity at an ambient temperature of 10˚C. For tomography of PAO1 Δ*pilT* cells, a bacterial lawn from an overnight LB agar plate incubated at 37˚C was resuspended in phosphate buffered saline (PBS), and 10 nm gold fiducials (CMC Utrecht) were added prior to plunge-freezing.

## Cryo-EM and cryo-ET data collection

Single particle cryo-EM data was collected on a Titan Krios G2 microscope (ThermoFisher) operating at an acceleration voltage of 300 kV, equipped with a Falcon 4i direct electron detector (ThermoFisher). Images were collected using a physical pixel size of 0.824 Å. For helical reconstruction of the T4P, movies were collected as 40 frames, with a total dose of 40 electrons/$Å^2$, using a range of nominal defoci between -1 and -2.5 µm. For the T4P single-particle dataset, 5,679 movies were collected. Tomography data of PAO1 Δ*pilT P. aeruginosa* cells was collected on a Titan Krios G3 microscope (ThermoFisher) operating at an acceleration voltage of 300 kV, fitted with a Quantum energy filter (slit width 20 eV) and a K3 direct electron detector (Gatan). Cryo-ET tilt series were collected using a grouped dose-symmetric tilt scheme as implemented in SerialEM [73], with a total dose of 175.5 electrons/$Å^2$ per tilt series, nominal defocus of -8 µm, and with ±60˚ tilts of the specimen stage at 1˚ tilt increments. Tilt series images were collected using a physical pixel size 3.42 Å.

## Cryo-EM processing

Helical reconstruction of the T4P was performed in RELION 4.0 [59,74–76] (S2 Fig). Movies were motion-corrected using the RELION 4.0 implementation of MotionCor2 [77], and CTF parameters were estimated using CTFFIND4 [78]. Initial helical symmetry of T4P used was obtained from previous reports in literature [26]. Three-dimensional (3D) classification was used to identify a subset of particles that supported refinement to 3.17 Å resolution. For final refinement, CTF multiplication was used for the final polished set of particles [57–59]. Symmetry searches were used during reconstruction, resulting in a final rise of 10.17 Å and a right-handed twist per subunit of 87.39˚. Resolution was estimated using the gold-standard Fourier Shell Correlation (FSC) method as implemented in RELION 4.0. Local resolution measurements (S3 Fig) were also performed using RELION 4.0.

## Model building and refinement

Manual model building of the PilA subunit was performed in Coot [79]. An AlphaFold2 [80] model was fitted into the cryo-EM density as a rigid body. Due to the more flexible linker region, the α-helical parts and β-sheet were initially treated separately. Residues of the

homology model that were inconsistent with the density were deleted and manually rebuilt. The initial model was subjected to real-space refinement against the cryo-EM map within the Phenix package [81,82]. Twenty-three subunits of PilA were built and used for final refinement. Non-crystallographic symmetry between individual PilA subunits was applied for all refinement runs. Model validation including map-vs-model resolution estimation was performed in Phenix (version 1.20).

## Tomogram reconstruction

Tilt series alignment via tracking of gold fiducials was performed using the IMOD [83] implementation in RELION 5 [84]. Tomograms were reconstructed in RELION 5 and denoised using CryoCARE [85] for visualisation purposes.

## Data visualisation and quantification

Atomic structures were visualised in ChimeraX [86]. Tomograms were visualised in IMOD [83]. Conservation analysis (Fig 4h) was performed using the ConSurf server [87–89]. Hydrophobicity (S6 and S11 Figs) and electrostatic (S10 Fig) plots were calculated in ChimeraX using the in-built *mlp* and *coulombic* functions. Alignment of structural models was performed using the *matchmaker* function in ChimeraX [86].

## Supporting information

**S1 Fig. Purification of *P. aeruginosa* PAO1 T4P. a** SDS-PAGE analysis of the final specimen after PEG precipitation (see Methods). Two bands, corresponding to FliC (flagella) and T4P PilA, are present at the expected molecular weights. The positions of molecular weight markers are shown on the left. **b** Peptide-fingerprinting mass spectrometry result of the T4P PilA band. The detected peptides are highlighted in the sequence, confirming the presence of T4P PilA in our sample.
(TIF)

**S2 Fig. Cryo-EM image processing workflow in RELION 4.0.** After particle picking and 2D classification (top 10 populated classes shown), the three best classes (114,537 particles, marked in red) were selected and used for initial 3D refinement yielding a 4.2 Å-resolution pilus structure. Following that, 3D classification was performed resulting in a final set of 76,998 particles, which were used for further 3D refinement. Further refinements and polishing were performed, resulting in the final 3.17 Å-resolution map.
(TIF)

**S3 Fig. Resolution estimation of the map. a** Cryo-EM density map coloured by local resolution (calculated in RELION 4.0). The slightly lower local resolution on the outside of the T4P is indicative of a higher degree of structural flexibility of the β-sheet domain compared to the α-helical part of the pilin. **b** Solvent-corrected masked half-map FSC (blue, gold-standard), phase-randomised half-map FSC (red), and model-vs-map FSC (black).
(TIF)

**S4 Fig. Side chain densities observed in the cryo-EM map. a-c** Models of the lower part of the N-terminal helix (residues 1–16, **a**), the 'melted' linker segment (residues 17–24, **b**) and the β-sheet (residues 85–130, **c**) fitted into the cryo-EM density. The side chains are clearly resolved in the map.
(TIF)

**S5 Fig. Comparison of the PAO1 T4P structure to the PAK T4P model. a** Sequence alignment [90] of *P. aeruginosa* pilA from strains PAO1 and PAK. **b** EM density map of the PAO1 T4P reported in this paper (EMD-50025), side view and front view of the EM density around one pilin subunit (PDB: 9EWX, this study) and EM density map of the PAK T4P reported in a previous study [26] (EMD-8740), along with side view and front view of the EM density around one pilin subunit (PDB: 5VXY). **c-f** Alignment of the pilin structures of PAO1 (dark blue) and PAK (teal), based on the globular domain (**c, e, f**) or the N-terminal helix (**d**), shows secondary structure differences, highlighting the relevance of a high-resolution map featuring side chain densities. Specifically, there is a difference in orientation between the N-terminal helix and the globular domain (**c, d**). In the globular domain, PAO1 features additional small secondary structure elements within the αβ-loop, while the loop between beta strands β1 and β2 is extended in PAK (red asterisks) (**e**). There are also differences in beta strand length between the two structures (red asterisk) (**f**).
(TIF)

**S6 Fig. Hydrophobic properties of the *P. aeruginosa* PAO1 Type IV pilin and pilus.** Hydrophobicity of the *P. aeruginosa* PilA subunit and the fully assembled T4P. The subunit is shown in two orientations related to each other by a 90˚ rotation (left). For the assembled pilus (right), both the outer surface and a cross section through the centre are shown. The models have been coloured by molecular lipophilicity potential using the *mlp* function in ChimeraX [86].
(TIF)

**S7 Fig. *P. aeruginosa* PAO1 T4P EM density (blue) modelled alongside the PilQ secretin density (EMD-8297) [35].** The secretin features a density in the lumen (top left) corresponding to the central gate, which needs to be displaced to accommodate the full pilus as described by Koo et al. [35]. The PAO1 T4P density fits well into the PilQ lumen.
(TIF)

**S8 Fig. Evolutionary conservation of T4Ps. a** Pilin structure coloured by ConSurf conservation score, where low numerical values (green) indicate low degrees and high numerical values (red) indicate high degrees of conservation. The analysis reveals a high degree of conservation of the N-terminal helix. A disulfide bond near the C-terminus (cysteine residues marked in pink) is also a feature conserved across many T4P species. **b** Full pilus structure coloured by ConSurf conservation score. **c** Sequence alignment and conservation score of the pilin from ConSurf analysis.
(TIF)

**S9 Fig. Comparison of the PAO1 PilA cryo-EM structure to the PAK PilA X-ray crystallography structure.** Side-by-side comparison of the PAO1 PilA (this study, left) with the PAK PilA (PDB: 1OQW, right) demonstrating the absence of the melted helical segment in the latter.
(TIF)

**S10 Fig. Comparison of surface electrostatics across T4P species. a-g** Models of T4P structures from different organisms coloured by electrostatic surface potential using the *coulombic* function in ChimeraX [86]: *P. aeruginosa* PAO1 (this study, PDB: 9EWX) (**a**), *E. coli* (PDB: 6GV9) (**b**), *N. gonorrhoeae* (PDB: 5VXX) (**c**), *T. thermophilus* (PDB: 6XXD) (**d**), *G. sulfurreducens* (PDB: 6VK9) (**e**), *M. xanthus* (PDB: 8TJ2) (**f**), *P. arsenaticum* (PDB: 6W8U) (**g**).
(TIF)

**S11 Fig. Comparison of hydrophobicity properties across T4P species. a-g** Models of T4P structures from different organisms coloured by molecular lipophilicity potential using the *mlp* function in ChimeraX [86]: *P. aeruginosa* PAO1 (this study, PDB: 9EWX) (**a**), *E. coli* (PDB: 6GV9) (**b**), *N. gonorrhoeae* (PDB: 5VXX) (**c**), *T. thermophilus* (PDB: 6XXD) (**d**), *G. sulfurreducens* (PDB: 6VK9) (**e**), *M. xanthus* (PDB: 8TJ2) (**f**), *P. arsenaticum* (PDB: 6W8U) (**g**). Negative potential corresponds to hydrophilic regions, positive potential to hydrophobic regions.
(TIF)

**S1 Table. Cryo-EM data acquisition and processing statistics.**
(DOCX)

**S2 Table. T4P from various organisms.**
(DOCX)

**S3 Table. RMSD values calculated between the *P. aeruginosa* PAO1 structure (9EWX) and other T4P species.** The values were calculated using two different methods: TM-align [91] and jFATCAT [92].
(DOCX)

**S1 Movie. Cryo-EM structure of the *P. aeruginosa* PAO1 T4P.** A 3.2 Å resolution cryo-EM density map of the T4P is shown, which was used to build an atomic model of the main pilus-forming subunit PilA (surface depiction and ribbon diagrams are shown).
(MP4)

**S1 File. Fasta-file of the sequence alignment performed while studying evolutionary conservation of pilins using ConSurf (S8 Fig).**
(PDF)

**S2 File. Full list of interacting residues within the PAO1 pilin.**
(DOCX)

## Acknowledgments

The authors would like to thank Prof. Urs Jenal and Dr. Benoit-Joseph Levantie for sharing the *pilT* deletion strain, and Andriko von Kügelgen for help with the cryo-EM data collection. We acknowledge the MRC-LMB electron microscopy facility for help with sample preparation and data collection and the MRC-LMB mass spectrometry facility for assistance with mass spectrometry analysis of the sample.

## Author Contributions

**Conceptualization:** Hannah Ochner, Tanmay A. M. Bharat.

**Data curation:** Hannah Ochner.

**Formal analysis:** Hannah Ochner, Jan Böhning, Zhexin Wang, Abul K. Tarafder, Ido Caspy, Tanmay A. M. Bharat.

**Investigation:** Hannah Ochner, Zhexin Wang, Abul K. Tarafder.

**Methodology:** Hannah Ochner, Zhexin Wang, Abul K. Tarafder.

**Project administration:** Tanmay A. M. Bharat.

**Supervision:** Tanmay A. M. Bharat.

**Validation:** Hannah Ochner, Jan Böhning, Tanmay A. M. Bharat.

**Visualization:** Hannah Ochner, Jan Böhning, Ido Caspy.

**Writing – original draft:** Hannah Ochner, Jan Böhning, Tanmay A. M. Bharat.

**Writing – review & editing:** Hannah Ochner, Jan Böhning, Zhexin Wang, Abul K. Tarafder, Tanmay A. M. Bharat.

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
