## [Decision Letter · Decision Letter 0]

11 Sep 2024

Dear Dr. Bharat

Thank you very much for submitting your manuscript "Structure of the Pseudomonas aeruginosa PAO1 Type IV pilus" for consideration at PLOS Pathogens. As with all papers reviewed by the journal, your manuscript was reviewed by members of the editorial board and by several independent reviewers. In light of the reviews (below this email), we would like to invite the resubmission of a significantly-revised version that takes into account the reviewers' comments.

The revised manuscript has been seen by two new reviewers and their comments are included. Both reviewers highlight the importance of the work. This revised manuscript addresses many of the comments raised previously. However, as pointed out by the Reviewer-1, there are still some issues that need to be resolved which can be addressed by rewriting the manuscript. I invite authors to submit the revised manuscript that addresses all the reviewer comments.

We cannot make any decision about publication until we have seen the revised manuscript and your response to the reviewers' comments. Your revised manuscript is also likely to be sent to reviewers for further evaluation.

Sincerely,

Nikhil Malvankar

Guest Editor

PLOS Pathogens

Matthew Wolfgang

Section Editor

PLOS Pathogens

Michael Malim

Editor-in-Chief

PLOS Pathogens

orcid.org/0000-0002-7699-2064

The revised manuscript has been seen by two new reviewers and their comments are included. Both reviewers highlight the importance of the work. This revised manuscript addresses many of the comments raised previously. However, as pointed out by the Reviewer-1, there are still some issues that need to be resolved which can be addressed by rewriting the manuscript. I invite authors to submit the revised manuscript that addresses all the reviewer comments.

Reviewer's Responses to Questions

**Part I - Summary**

Reviewer #1: The manuscript “Structure of the Pseudomonas aeruginosa PAO1 Type IV pilus” presents the high quality cryoEM reconstruction of the Type IV pilus (T4P) from an important human pathogen. This structure represents a clear advance from the previously published 8 Å T4P structure from P. aeruginosa strain K (PAK, ref. 26). The higher resolution allows the entire pilin chain to be fit. For the lower resolution structure reported previously only the N-terminal helix was resolved and the filament structure was obtained by fitting in the x-ray crystal structure of the PAK pilin subunit. Nonetheless, and despite the two pilins sharing only 67% identity, the two structures are highly similar, and this is never acknowledged in the current manuscript, which presents this T4P structure as completely new and novel. For instance, the abstract states that “PilA subunits constituting the T4P exhibit a classical pilin fold featuring an extended N-terminal a-helix linked to a C-terminal globular b-sheet-containing domain, which are packed tightly along the pilus. The N-terminal helices constitute the pilus core where they stabilise the tubular assembly via hydrophobic interactions. The a-helical core of the pilus is surrounded by the C-terminal globular domain of PilA that coats the outer surface of the pilus, mediating interactions with the surrounding environment.” All of this was described for the lower resolution structure. In the first section of results (indeed the only results presented) the entire description they provide for the PAK pilus structure was described previously (ref. 26 for PAK pilus, and a multitude of papers for other T4P). The “key differences” and “common architectural principles” that this new structure apparently reveals are well-established. Thus, the findings are confirmatory, not novel, and while this high resolution structure is of value it’s not clear what new information it provides beyond confirming the accuracy of the 8 Å structure. The manuscript should be reframed: present a side-by-side comparison of the two structures and highlight differences, if there are any.

The authors provide some global comparisons of T4P structures and highlight some differences and speculate on their significance but this is a fairly superficial analysis. For example, in lines 177-180, they say that differences in the N-terminal a-helix position influence the pilus architecture but do not explain this statement. How do they influence the pilus architecture? In lines 194-201 they speculate that the lack of surface-displayed loops makes P. aeruginosa T4P more resistant to proteolysis but provide no empirical data to support this. In line 205-208 they discuss the surface electrostatics and suggest that the positively charged regions may correlate with DNA binding regions but provide no analysis on which T4P are actually involved in natural transformation. (Note that the major pilin is not necessarily the site of DNA binding.)

Other points:

Line 107-151: The entire description of the T4P structure was described previously for the 8 Å PAK pilus structure. This section should start with a description of the map and compare it with 8 Å map to highlight the structural advance. Provide a sequence and structural comparison for the two T4P structures (and cite the recently published 3.6 Å PAK pilus structure, ref. 68), including of the pilin subunits. How do they differ? What is learned from this higher resolution structure.

Line 113-114: “Next, we used a previous estimate of the helical symmetry of the P. aeruginosa

PAK strain T4P …”. This was not an “estimate”, it was an empirically determined value.

Line 136: What is meant by “hydrophobic stacking”? This is a staggered arrangement, not a stacking.

Line138: “residues 7-22” Residues are numbered based on the pre-pilin. The convention for T4 pilins is to number the residues beginning at the mature protein. Thus, this should be “residues 1-16”.

Reviewer #2: This is a revised submission of a previously reviewed paper on the 3.2 Å cryo EM structure of the P. aeruginosa PAO1 type IV pilin structure. The structure is well-executed and the experimental methodology is sufficiently explained. Although there is a contemporaneous publication of a 3.6 Å structure of the same filament, this is still a useful publication at slightly higher resolution that verifies and expands those results.

The authors seem to have made significant changes in response to the previous review.

**Part II – Major Issues: Key Experiments Required for Acceptance**

Reviewer #1: (No Response)

Reviewer #2: N/A

**Part III – Minor Issues: Editorial and Data Presentation Modifications**

Reviewer #1: (No Response)

Reviewer #2: I have only one comment to address.

1. On page 163, and in Fig. S9, the crystal and EM structures are compared and the “melted” region in the EM studies is noted as a difference. Authors should clarify it’s not more or less accurate to have the unraveled helix than the helical section in the crystal structures, and it is not the case that different bacterial pilins would have different structures here, nor that the resolution plays any role in the differences. Rather, the crystal structures are determined of pilin subunits in biological detergents, and the alpha-1 helices in these cases have relaxed to the lowest free energy structure, the helix. On the other hand, in assembled filaments, the unraveled helix is maintained.

PLOS authors have the option to publish the peer review history of their article (what does this mean?). If published, this will include your full peer review and any attached files.

Reviewer #1: No

Reviewer #2: No
---

## [Decision Letter · Decision Letter 1]

25 Nov 2024

Dear Dr. Bharat

We are pleased to inform you that your manuscript 'Structure of the Pseudomonas aeruginosa PAO1 Type IV pilus' has been provisionally accepted for publication in PLOS Pathogens.

Best regards,

Nikhil Malvankar

Guest Editor

PLOS Pathogens

Matthew Wolfgang

Section Editor

PLOS Pathogens

Michael Malim

Editor-in-Chief

PLOS Pathogens

orcid.org/0000-0002-7699-2064

Reviewer Comments (if any, and for reference):

Reviewer's Responses to Questions

**Part I - Summary**

Reviewer #1: The authors have address my concerns.

Reviewer #2: I don't have anything else to add.

**Part II – Major Issues: Key Experiments Required for Acceptance**

Reviewer #1: (No Response)

Reviewer #2: (No Response)

**Part III – Minor Issues: Editorial and Data Presentation Modifications**

Reviewer #1: (No Response)

Reviewer #2: (No Response)

PLOS authors have the option to publish the peer review history of their article (what does this mean?). If published, this will include your full peer review and any attached files.

Reviewer #1: No

Reviewer #2: No

---

## [Editor Report · Acceptance letter]

2 Dec 2024

Dear Dr Bharat,

We are delighted to inform you that your manuscript, "Structure of the Pseudomonas aeruginosa PAO1 Type IV pilus," has been formally accepted for publication in PLOS Pathogens.

Best regards,

Michael Malim

Editor-in-Chief

PLOS Pathogens

orcid.org/0000-0002-7699-2064